# An External Energy Independent WO$_3$/MoCl$_5$ Nano-Sized Catalyst for the Superior Degradation of Crystal Violet and Rhodamine B Dye

**Dongin Kim, Geonwoo Kim, Hyeonbin Bae, Eunwoo Kim, Byunghyun Moon, Daho Cheon and Naresh Hiralal Tarte \***

Department of Chemistry and Biology, KSA of KAIST, 105-47, Baegyanggwanmun-ro, Busanjin-gu, Busan 47162, Korea

\* Correspondence: naresh@kaist.edu; Tel.: +82-10-5060-0598

**Abstract:** In this study, the synthesis of a novel catalyst WO$_3$/MoCl$_5$ was carried out by the thermal method. The method gave an entirely different product compared to previous studies that doped Mo on the surface of semiconductor metal oxides. The degradation reaction of crystal violet (CV) and rhodamine B (RB) dye were done without any energy source. The results showed an incomparably superior result for degradation, with a reaction rate constant of 1.74 s$^{-1}$ for 30 ppm CV, 1.08 s$^{-1}$ for 30 ppm RB, and a higher value than 1 s$^{-1}$ for both cases of 50 ppm dye solution. To the author's knowledge, this catalyst has the highest reaction rate compared to other studies that targeted CV and RB, with an immense reaction rate increase of more than 100 times. Reusability of the three trials was verified, and the only process required was washing the catalyst after the reaction. One of the drawbacks of the advanced oxidation process (AOP), which has a degradation percent limit, has been solved, since 100% mineralization of the dye was available using this catalyst. FT-IR spectroscopy revealed that W-O-Mo linkage was successfully processed while Mo-Cl linkage has retained. $^1$H-NMR spectroscopy results confirmed that the degradation product of the dye treated by simple MoCl$_5$ and WO$_3$/MoCl$_5$ was different. Deep inspection of specific regions of NMR fields gave necessary information about the degradation product using WO$_3$/MoCl$_5$.

**Keywords:** semiconductor metal oxide; MoCl$_5$; novel thermal synthesis pathway; crystal violet dye; rhodamine b dye; dye waste treatment

---

## 1. Introduction

Due to the heavy usage of organic chemicals, the extensive industrial spoilage has occurred. Throughout aquatic environments. Since a noticeable increase in the disposal of refractory organic materials has the potential for environmental and biological damage and has already severely impacted water quality [1], a novel technology for water treatment has gained significant interest. Dyes in wastewater are proposed to be pollutants that cause considerable environmental damage, including toxicological effects on micro-organisms and color pollution [2–4]. Specifically, rhodamine B (RB), which is one of the target compounds in this study, is a dye extensively used in various industries, such as textiles, leather, and food [5]. Nevertheless, RB is also a well-known carcinogen and irritant for the skin, eyes, and respiratory tract. It also possesses reproductive and developmental toxicity and neurotoxicity in humans and animals [6–9]. Further, crystal violet (CV) has been widely used in veterinary and human medicine as a biological stain, as well as a mutagenic and bacteriostatic agent in medical solutions [10], and is known to have genotoxicity and functions as a carcinogen [11–13]. Therefore, various researchers have focused on the development of a method to remove this organic

pollutant from the aqueous phase. Adsorption, oxidation, reduction, electrochemical reactions [14–16], and many other methods have been considered. Especially in recent years, Advanced Oxidation Processes (AOPs) have been verified to be very promising for organic degradation [17–19].

As one of the most effective and widespread AOPs, the Fenton process has distinctive advantages due to its generation and usage of an active hydroxyl radical by simple redox reactions between $H_2O_2$ and Fe(II). The Fenton process also requires a low cost and has few environmental risks [20]. Most organic compounds can be degraded into less toxic compounds under ambient temperature and pressure conditions [20,21]. However, due to the difficulty of reusability caused by homogeneous features, a heterogeneous Fenton-like process has been investigated instead of the Fenton process and Fenton-related processes. For instance, iron-free catalysts, including gold [22], copper [23], and manganese [24], were developed for a heterogeneous Fenton-like process in the presence of $H_2O_2$. However, such AOPs require additional supplies, such as $H_2O_2$, ozone, or an external energy source and need a relatively long time for catalysis. Therefore, a novel mechanism for organic pollutants should be developed to enhance wastewater treatment methods.

$WO_3$ has been widely used as a visible light-response photocatalyst absorbing light at wavelengths up to 480 nm and has intriguing advantages, such as its harmless low cost and stability [25–28]. $MoCl_5$ is an aggressive oxidant but cannot be recycled [29,30]. Therefore, by combining the semiconductor $WO_3$ and the oxidant, a recyclable oxidant can be constructed.

Herein, a $WO_3/MoCl_5$ catalyst was prepared by thermal synthesis and assessed for the oxidative degradation of Crystal Violet (CV) and Rhodamine B (RB) in the aqueous phase. The characterizations of synthesized materials were done using FT-IR, XRD, EDS-SEM, and TEM which all justify the successful linkage between $MoCl_5$ and $WO_3$ and the conservation of basic structures of the metal oxides during synthesis and also after reaction. NMR analysis of degradation products using pristine $MoCl_5$ and the synthesized material also legitimized the difference between simple mixture and the catalysts. The synthesized catalyst showed a prominent degradation efficiency and an incomparably short-term degradation compared to previous studies, without any additional supplies or external energy sources. Additionally, we conducted a series of experiments and analyses on degradation kinetics, such as dark reaction and a radical scavenger reaction, which included reaction and recycling.

## 2. Results

### 2.1. Morphological and Microstructural Analysis

The morphology of the precursors and $MoCl_5$ linked catalysts was shown by the TEM images in Figure 1. The particle sizes of the precursor and $WO_3/MoCl_5$ had high randomness in both shape and size. However, the metal precursor had a greater deviation of size and was typically bigger than the synthesized catalyst. The new particle sizes, which are all in the range of nanoparticles, ranged from 60 to 90 nm. The synthesized Mo-doped catalyst had more blurred and uncertain boundaries than the pristine precursor.

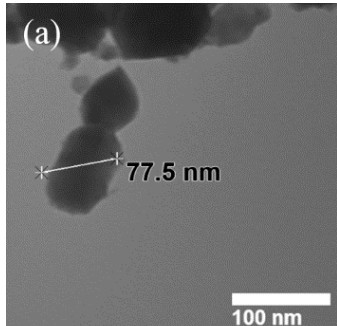 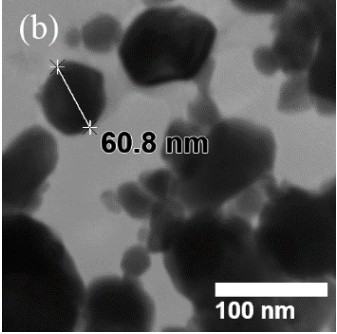

**Figure 1.** TEM images of the precursor and $MoCl_5$ linked catalyst: (**a**) $WO_3$; (**b**) $WO_3/MoCl_5$.

Figure 2 and Table 1 show the result of the EDS analysis of the synthesized catalyst. Figure 2 shows the reflected characteristic K$\alpha$ signals of specific elements, which were used to assign each element deposited on the catalyst surface. Table 1 shows the element proportion of the catalyst on the surface, which was usually composed of Tungsten, Chlorine, Oxygen, and Molybdenum. The coexistence of Mo and Cl on the surface confirmed that $MoCl_5$ was linked to Tungsten oxide. To exclude the probability of small remnants of $MoCl_5$ effecting the EDS result, the synthesized catalysts were washed with toluene until the proportion of Mo and Cl reversed. A high percentage of nitrogen on the surface was due to adsorbed $N_2$ gas. The absence of carbon in the surface assures that there were no remaining residuals, such as 1,1,3,3-Tetramethylguanidine (TMG) and other precursors. TMG was used in the synthesizing procedure to deprotonate the $WO_3$ precursor. The EDS analysis result shows that the precursor had been totally washed away by toluene. In summary, $MoCl_5$ successfully made a conjunction with tungsten oxide.

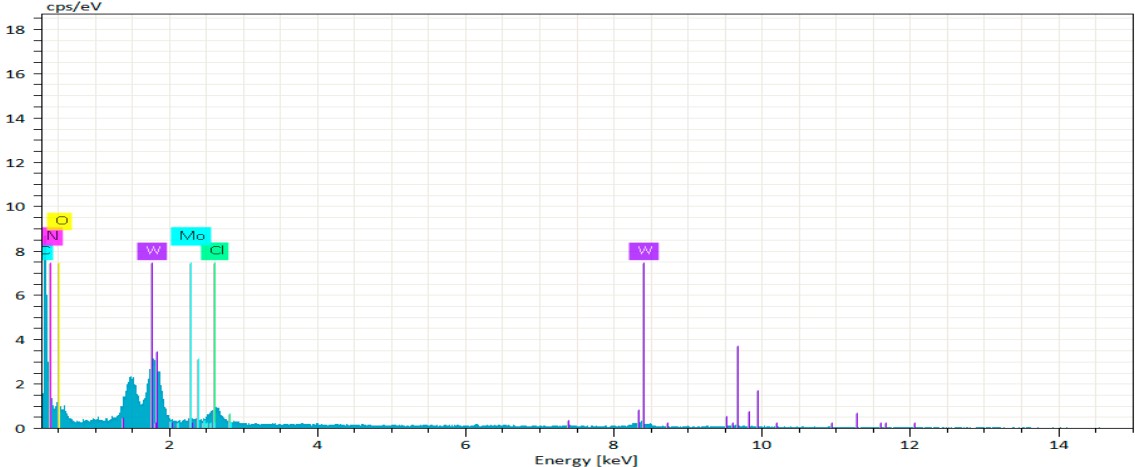

**Figure 2.** Element assignment by EDS analysis of $WO_3/MoCl_5$.

**Table 1.** Element proportion analysis of $WO_3$-Mo.

| Element | Atom No. | Mass Norm [%] | Atom [%] | Abs. Error [%] |
|---|---|---|---|---|
| Nitrogen | 7 | 52.3 | 72.17 | 19.38 |
| Chlorine | 17 | 3.97 | 2.16 | 0.52 |
| Oxygen | 8 | 18.97 | 22.92 | 8.62 |
| Carbon | 6 | 0 | 0 | 0 |
| Tungsten | 74 | 23.24 | 2.45 | 3.1 |
| Molybdenum | 42 | 1.52 | 0.31 | 0.31 |

### 2.2. Structural Analysis

The XRD pattern of $WO_3$ was assigned via JCPDS No. 75-2072. At $2\theta = 17 \sim 19°$, the calculated characteristic peak of the orthorhombic structure of $MoCl_5$ newly appears, indicating that the total $MoCl_5$ is linked to the original structure of the metal precursors. This result matches well with the EDS result, showing the existence of Mo and Cl on the surface of the catalyst. Peak broadening and intensity decreasing were induced by the $MoCl_5$ linking on the crystalline structure of $WO_3$.

The infrared spectra of $WO_3$ and $WO_3/MoCl_5$ are shown in Figure 3. The infrared spectra of the previously synthesized and published $ZrO_2/MoCl_5$ and $SrTiO_3/MoCl_5$ particles are compared in Figure 3 with $WO_3/MoCl_5$ spectra to support the IR peak assignation. For $WO_3$, the broad peak observed in the regions of 3600–3400 cm$^{-1}$ and 1600 cm$^{-1}$ are attributable to the presence of hydroxyl groups bonded to the metal at the surface [31]. These signals are no longer visible at the spectra of the synthesized catalysts, which implies that the hydroxyl groups have first been deprotonated by the basicity of TMG and then formed a new Mo-O-W bond during the synthesis. A peak at 975 cm$^{-1}$ and

weak signals at 565 cm$^{-1}$ and 450 cm$^{-1}$ are suspected to be evidence for the successful MoCl$_5$ linkage. The primary peak (975 cm$^{-1}$) indicates the stretching vibration mode of the bond of Mo-O-Metal [32]. The rest signals result from the Mo-O bond (565 cm$^{-1}$) and Mo-Cl bond (450 cm$^{-1}$), respectively [32,33]. These analyses rationalize the formation of the bond between Mo and O and the remaining of Mo-Cl bond, which is conclusive evidence that Molybdenum linked to the metal precursor, retaining its linkage to Cl. All of the peaks are common among WO$_3$/MoCl$_5$, ZrO$_2$/MoCl$_5$, and SrTiO$_3$/MoCl$_5$, which suggests that the assignments are replicable and, therefore, plausible.

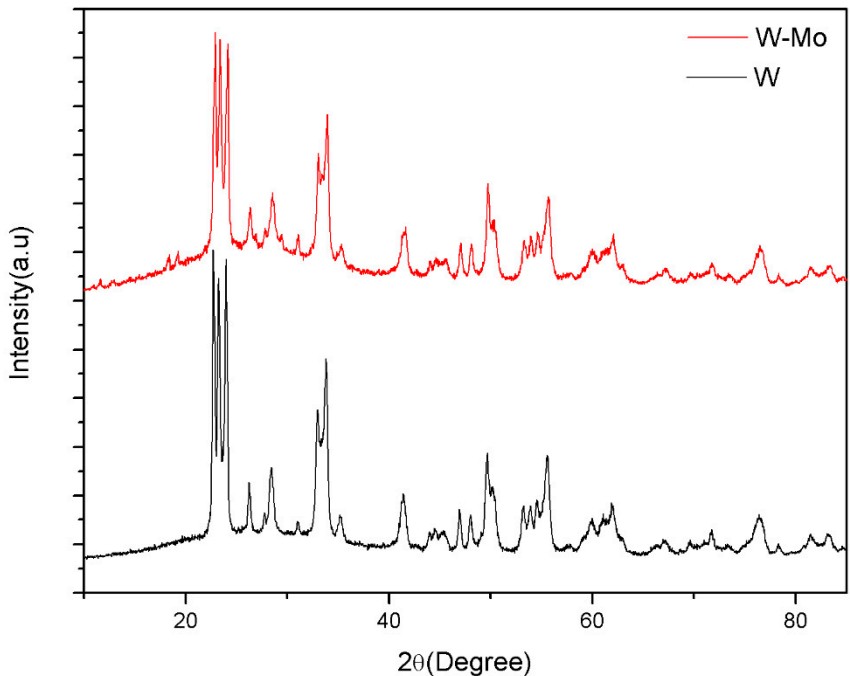

**Figure 3.** The XRD pattern of the precursor WO$_3$ and WO$_3$-Mo.

Other newly formed familiar peaks of the synthesized catalysts are broad bands at the 3000–3400 cm$^{-1}$ region, with noticeable peaks at around 1600, 1032, 1065, and 1083 cm$^{-1}$ (Figure 4). These peaks are assigned to the adsorbed TMG FT-IR spectrum via the SDBS No. 10085 file. The existence of TMG peaks is acceptable because the synthesized catalyst was not washed by toluene, unless it produces new peaks or shifts preexisting peaks. The specified signals of the spectra are assigned for each characteristic vibration by a comparison between the precursor and the Mo-doped metal oxide. Standard signals in the WO$_3$ and WO$_3$/MoCl$_5$ ranges at 1048 and 984 cm$^{-1}$ correspond to the W-O and W=O bonds, respectively [34]. The natural metal oxide and MoCl$_5$ linked metal oxide showed identical signals with respect of the characteristic W-O bond. This result substantiates that the doping of Molybdenum occurred without modifying the structures of the precursors.

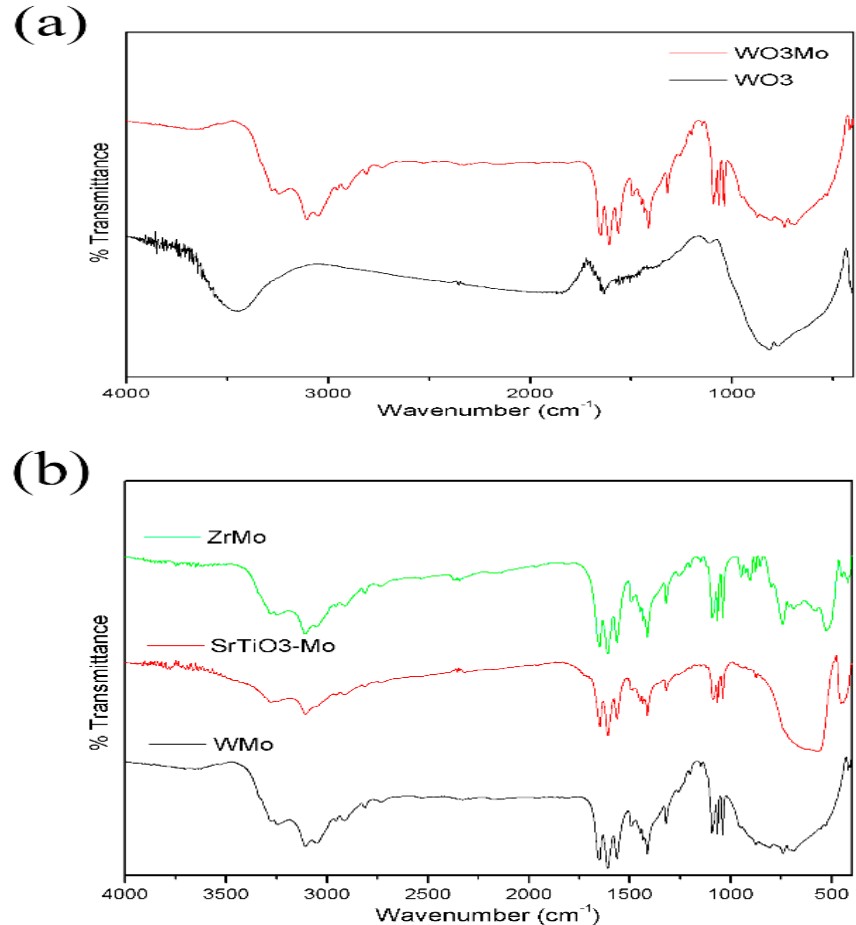

**Figure 4.** (**a**) FT-IR spectra of WO$_3$ and Mo doped WO$_3$ (**b**) Comparison of FT-IR peaks of Mo doped metal oxides.

## 2.3. Catalytic Study

Figure 5 shows the catalytic results of CV and RB using synthesized catalysts under normal light conditions. The only light energy that the reaction could gain was from a fluorescent lamp located at the ceiling to brighten the laboratory. This light is almost negligible.

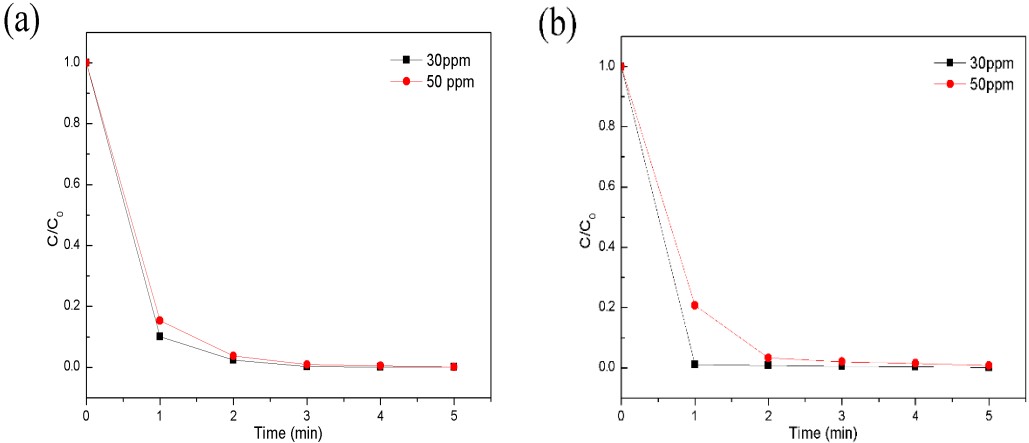

**Figure 5.** (**a**) Degradation of Crystal Violet (CV), 30 ppm (black) and 50 ppm (red); (**b**) degradation of Rhodamine B, 30 ppm (black) and 50 ppm (red).

The reaction was conducted to degrade 30 ppm and 50 ppm of the dyes, which is a high concentration compared to other studies about CV and RB degradation [35–41]. Trials with 10 ppm and 20 ppm were unable to be analyzed since all of the degradations were done within a minute, and observing the concentration for each shorter time interval was too risky from the perspective of accuracy. Even though 30 ppm and 50 ppm of dyes required three to five minutes to degrade fully, at least 80% of the dyes were degraded within a minute for all cases. The reaction rate constant was 1.74/s (104.4/min) in the case of the 30 ppm crystal violet, which is the highest to the author's knowledge. RB degrading studies done by Yao W. et al., Wilke K. et al., and Wei B.X. et al. showed a degradation rate constant of 1.3/min using $Ag_3PO_4/TiO_2$ nanoparticles, 1.1/min using $Cr^{3+}$ doped $TiO_2$, and 0.6/min using $ZrO_2$-coated $TiO_2$, respectively [35–37,41]. CV degrading studies done by Sahoo C. et al., Ameen, S. et al., and Sajid, M. et al. showed a degradation rate constant of 0.83/min using Ag doped $TiO_2$, 1.25/min using flower shaped ZnO nanoparticles, and 1.12/min using $BiVO_4/FeVO_4$ nanoparticles, respectively [38–40]. Other degradation coefficients of this study are summarized in Table 2. One of the drawbacks of AOPs was that the majority of the case reached their equilibrium limit in degradation, and only a rare case of treatment could give total degradation. Examples of degradation reactions reaching their equilibrium limits include studies done by Catala et al. (degrading mixture of drugs, 70% removal, 6 h using silica supported iron nanoparticles), Chong et al. (degrading Diclofenac 20 ppm, 85.2% removal, 40 min using Fe doped $CeO_2$), and Jaafarzadeh et al. (degrading Tetracycline 20 pppm, 44.8% removal, 180 min using nano carbon and $Fe_3O_4$) [42–44]. On the other hand, examples of total degradation include studies done by Arzate et al. (100% degradation of 30 ppm diclofenac after 90 min using an $Fe_2O_3$ catalyst with 500 $W/m^2$ light intensity) and Bobu et al. (100% degradation of 0.15 mM ciprofloxacin after 30 min using Fe-clay nanocomposites with a 125 W lamp) [45,46]. However, the newly synthesized catalysts were both extremely fast and could eliminate the dye in solution.

**Table 2.** First order kinetics of dye degradation reaction (RB and CV; normal (light) and dark condition).

| Dye | CV 30 | | CV 50 | | RB 30 | | RB 50 | |
|---|---|---|---|---|---|---|---|---|
| | $R^2$ | k [$sec^{-1}$] | $R^2$ | k [$sec^{-1}$] | $R^2$ | k [$sec^{-1}$] | $R^2$ | k [$sec^{-1}$] |
| Normal Condition | 0.986 | 1.744 | 0.990 | 1.178 | 0.737 | 1.08 | 0.898 | 0.928 |
| Dark Condition | 0.961 | 1.378 | 0.984 | 1.363 | 0.932 | 1.218 | 0.986 | 0.884 |

Comparative studies on the perspective of catalytic activity were done with $WO_3$ nanoparticles to emphasize the change after linking $MoCl_5$ to a pristine metal oxide. A 100 mJ laser pulse induced degradation of C.I. Acid Red 87 (25 ppm), using $WO_3$ catalysts, in a study Qamar M. et al., which showed a 20/min degradation rate [47]. Another laser induced degradation of Safranin-O (20 ppm), using nanocrystalline $WO_3$, was done by Hayat K. et al., who a showed 10/min degradation rate when the laser was used, and showed a 1.2 min degradation rate when no other energy was given to the reaction [48]. Other studies using only natural $WO_3$ as a catalyst without an energy source were unable to be found, which is natural since $WO_3$ is a semiconductor, which needs external energy to excite their electrons. This comparative study with pure $WO_3$ and $WO_3/MoCl_5$ restated the effect of linking $MoCl_5$ to $WO_3$, especially considering other studies that showed lower reaction rates than the reaction rate of the new catalyst of this study, even after using high energy laser sources.

After being added to the catalysts, the dye solution showed an instant color change for both dyes. In the case of Rhodamine B, the original pink color changed to purple, and the purple crystal violet solution changed to yellow. Related to the fact that the crystal violet changed its color in acidic condition, we used a pH meter to check the pH of each solution after adding the catalyst. The pH value was measured straightaway after adding the catalyst to the dye solution, while the solution color remained purple and yellow for RB and CV, respectively. Both of the solutions had a pH value of 0.5–1.0, which is extremely acidic. This result suggests that new material different from separate $MoCl_5$ and $WO_3$ was synthesized. A color change of RB, which is not an indicator, may be attributed

to a difference in the optical properties of the pure RB and the RB-catalyst complex. After 1 min, the color of the solution disappeared totally, as all of the dyes were degraded into colorless molecules.

To preclude the effect of the dim light extant when the normal light reaction occurred, dark condition degradation was also conducted, and the results are shown in Figure 6.

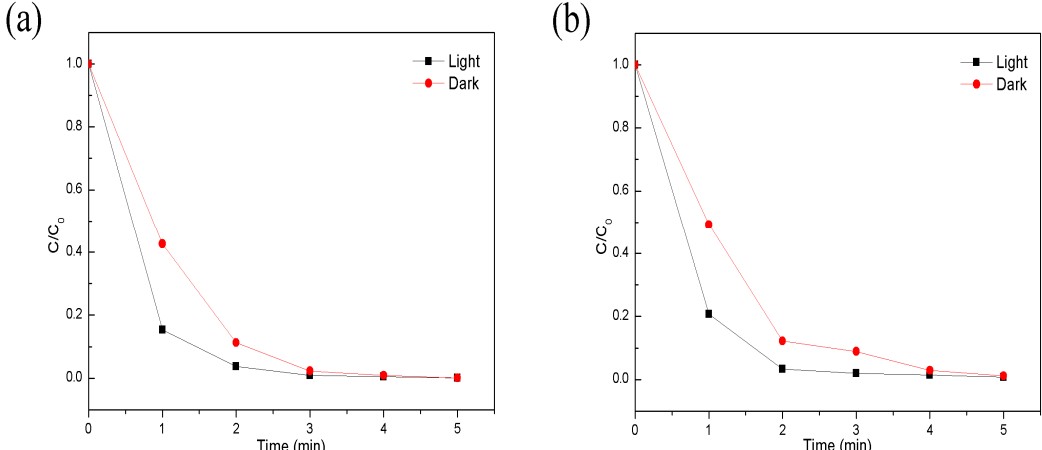

**Figure 6.** (**a**) Degradation of Crystal Violet (CV) 50 ppm at ordinary light conditions (black) and a totally dark condition (red); (**b**) degradation of Rhodamine B 50 ppm at ordinary light conditions (black) and a totally dark condition (red).

Analyzing the effect of light was done by comparing a 50 ppm concentration solution degradation of both CV and RB. The results showed that more time was needed for the degradation of dyes in the dark condition. Furthermore, the dyes in the dark condition had smaller reaction rate constants compared to those in the normal light condition. Nevertheless, the dye solution was completely decolorized in the dark dark condition after 5 min. This result shows that energy sources in this reaction might provide a difference from kinetic perspective but not from a thermodynamical one, at least for the 50 ppm dye. Additionally, this lag in the time required for total degradation did not affect the prestige of the catalyst, since 5 min is, relatively, 12–30 times shorter than the time for the other catalysts [35–41].

Table 2 presents pseudo first-order kinetics of the degradation of RB and CV at the normal light condition and the dark condition. The first-order kinetics were calculated by using the equation, $-\ln\left(\frac{C}{C_0}\right) = kt$. $C$, $C_0$, $k$, and $t$ represent each concentration at each time, the original concentration, the reaction rate in $s^{-1}$ units, and the corresponding time, respectively. The respective reaction rate constants ($k$ [$s^1$]) and the determination coefficients ($R^2$) have been listed. The reaction rate constants were measured in $s^{-1}$ since they almost exceeded 100 in $min^{-1}$. Three trials were done with each reaction, showing repetition, with a value of uncertainty of 0.0007 in the maximum on reaction rate values. All values in Table 2 are an average of the three trials, and the values are only written to three digits after the period due to uncertainty. Compared to many other previous studies [35–41], the synthesized catalyst had eminently high reaction rate constants. Except for the CV 30 ppm degradation, all of the regression coefficients had values that were high enough to demonstrate that each reaction occurred according to first-order kinetics. The assumption that a blank solution would have a value of 0.001 might have been a major factor for tge regression coefficient of CV 30 ppm, with the normal light condition having relatively low value. The regression coefficient calculated from CV, 30 ppm (normal light condition), excluding the fourth and fifth minute, showed 0.9372, which is justifiably assumed to possess first-order kinetics.

Although various analysis of the catalyst suggest that new catalysts different from crude precursors were synthesized, it is still disputable if the degradation process results from residual MoCl$_5$ or the catalyst. Additionally, it should be verified if the process is degradation or simple adsorption.

Figure 7; Figure 8, which are both $^1$H-NMR spectroscopy data, suggest a conclusive result for each problem, respectively.

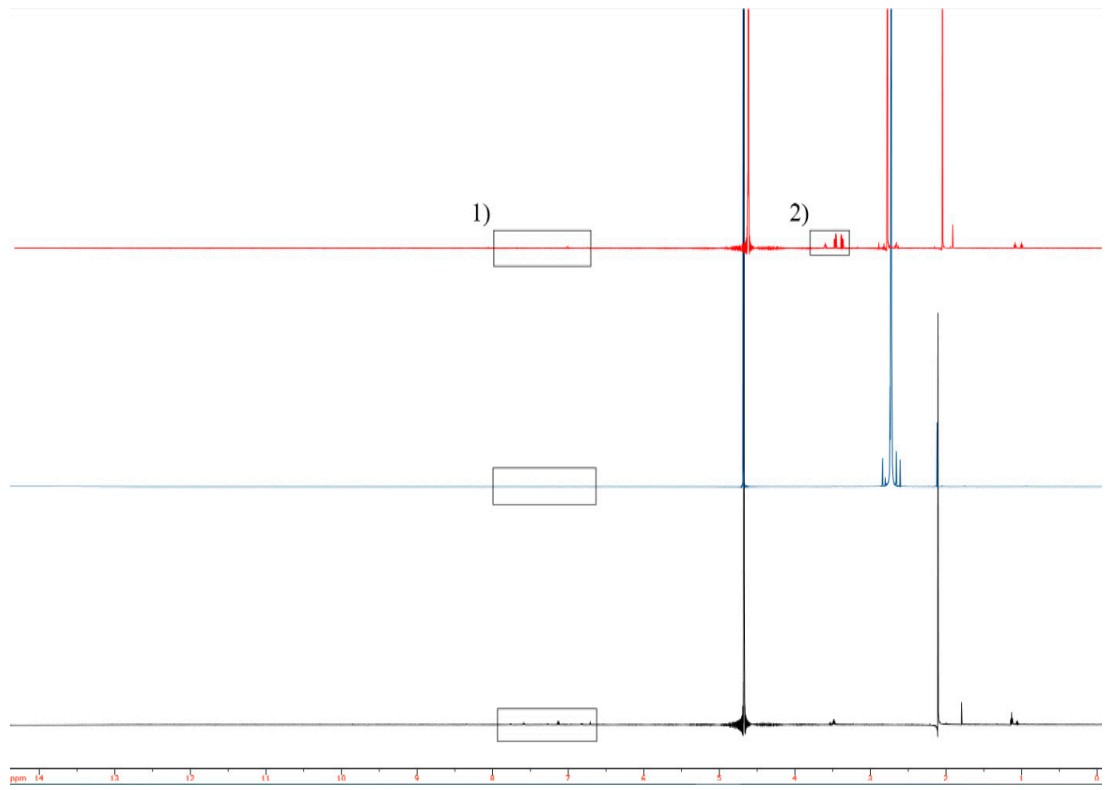

**Figure 7.** 600 MHz $^1$H-NMR peaks of rhodamine B (RB) (black), the degradation product of RB by MoCl$_5$ (blue), and the degradation product of RB by the synthesized catalyst (red).

Figure 7 shows 600 MHz quality $^1$H-NMR spectroscopy data at full scale for $\delta = 0 - 14$ ppm. To compare the degradation product of MoCl$_5$ and WO$_3$/MoCl$_5$, all data were merged with the data of RB $^1$H-NMR and are displayed in one figure. Based on Figure 7, the degradation peak of MoCl$_5$ and WO$_3$/MoCl$_5$ showed a noticeable difference. In the box labeled as 1), it is observable that the original peak of RB in the region of $\delta = 6.5 - 8$ ppm remains after being treated by WO$_3$/MoCl$_5$, whereas the RB solution treated by MoCl$_5$ is deficient with the corresponding peak. Besides the 1) box, the region is slightly upfield from 3 ppm (a broad peak at $\delta = 2.5 - 3$ ppm) and $\delta = 2 - 2.2$ ppm; $\delta = 1.8 - 2.0$ ppm has a different peak appearance. Among all characteristic peak differences, the alteration that occurred at the box labeled as 2) is the most noteworthy, since a new peak appeared, and this peak exactly matches previous research on the degradation of RB conducted by Steve D. et al. [49]. This is key evidence that WO$_3$/MoCl$_5$ follows the process of degradation that is already well studied and different from that of pure MoCl$_5$. To give a detailed comparison of the $^1$H-NMR peak of RB and RB treated with WO$_3$/MoCl$_5$, a magnified image at a specific region with a size of 1 ppm is shown (Figure 8).

Each of the comparisons of a specific region of the 600 MHz $^1$H-NMR peak shows the same tendency, in which some of the peaks remain, and other peaks are either shifted or eliminated. Several new peaks became visible, too (Figure 8a–c).

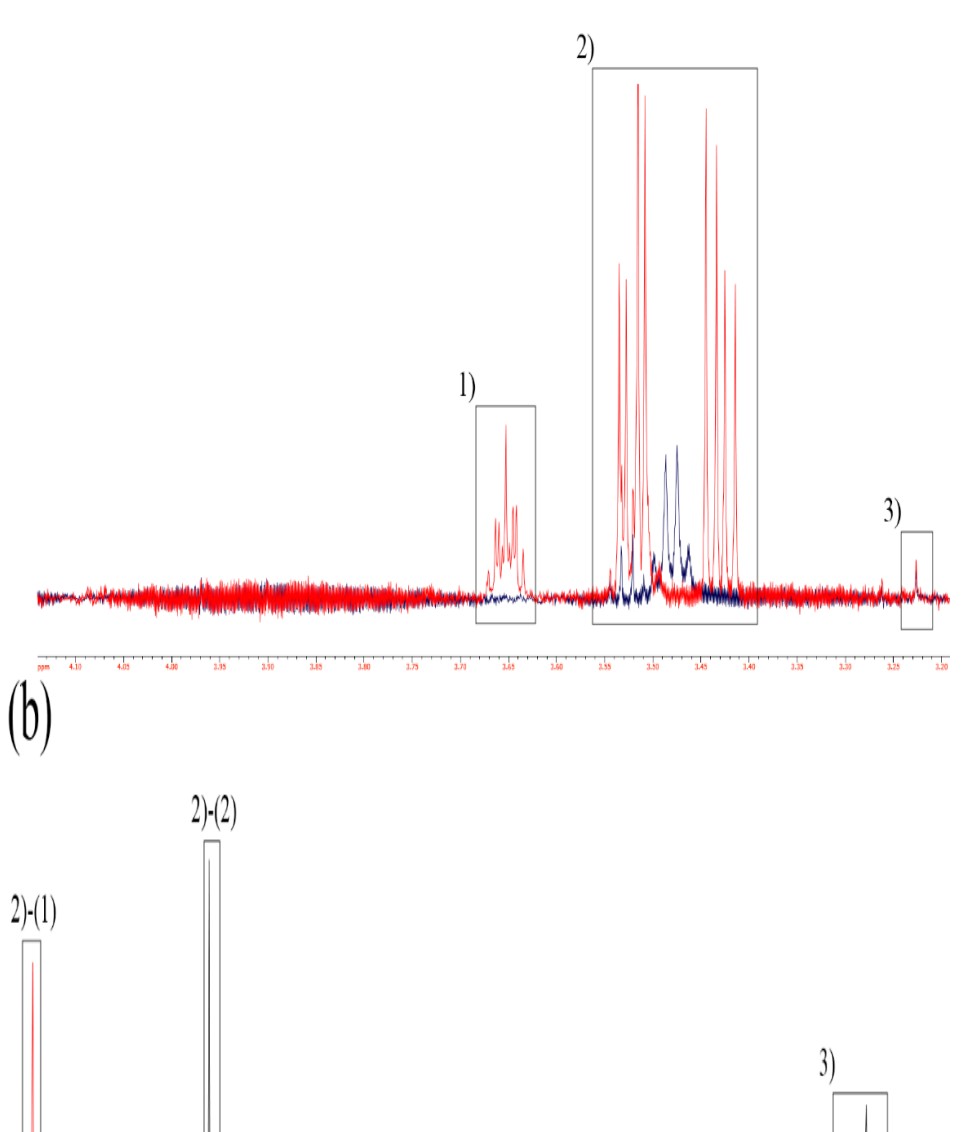

**Figure 8.** *Cont.*

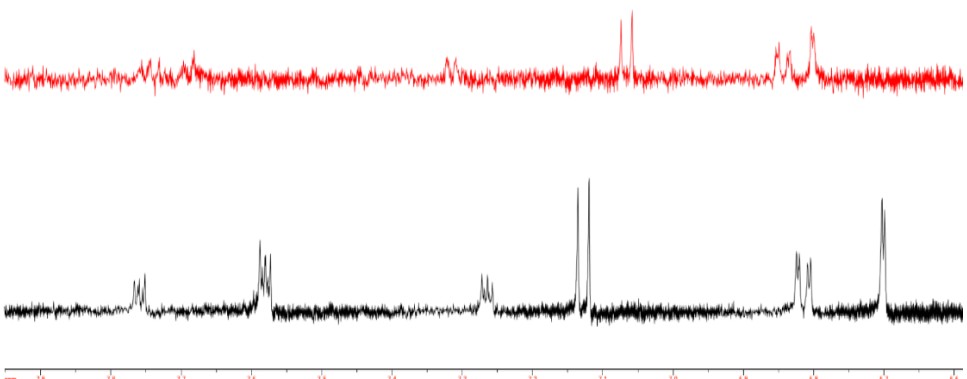

**Figure 8.** 600 MHz $^1$H-NMR characteristic peak comparison of the degradation product of Rhodamine B by synthesized catalyst (red) and RB (black), focusing on the (**a**) $\delta = 3 - 4$ ppm, (**b**) $\delta = 1 - 2$ ppm, and (**c**) $\delta = 6.5 - 8$ ppm shifts.

$\delta = 3 - 4$ ppm represents hydrogen in RO-C$^1$H and NC-C$^1$H, and the degradation results show (Figure 8a) that the peak inside the box labeled as 3) remains, while the peak inside the box labeled as 1) newly appeared. The peak in box 2) has either divided into two from the initial peak or the original peak might have disappeared, while two new peaks appeared. The peaks in the box labeled 2) suggest the unsymmetrical cracking of the RB molecule into two fragments both including NC-C$^1$H but with a different shape (Figure 8a).

Saturated alkane linked hydrogens appear in the range of $\delta = 1 - 2$ ppm (Figure 8b). Peaks included in the boxes labeled as 1) and 4) remained, and the peaks in 3) shifted. The peaks in 2)-(1) and 2)-(2) are both singlets, and the relationship between two peaks might be that 2)-(2) shifted to 2)-(1), or 2)-(1) could be an entirely different peak.

Aromatic hydrogens appear in the region of $\delta = 6.5 - 8$ ppm (Figure 8c). The included peaks were shifted or eliminated as WO$_3$/MoCl$_5$ was treated. All of the results by $^1$H-NMR eliminated the probability of simple adsorption, since the same RB solution with a different concentration only shows a difference in peak intensity and not in peak allocation and shape. Summarizing all of the $^1$H-NMR results in Figure 7; Figure 8, synthesized WO$_3$/MoCl$_5$ followed the degradation pathway, and the degradation products seemed to be different. Notably, a full-scale comparison ($\delta = 0 - 14$ ppm) of $^1$H-NMR shows that the $^1$H-NMR peaks matched up with other studies on RB degradation [49], and peaks at $\delta = 3 - 4$ ppm shows that the degradation product might have resulted from the two molecules cracking unsymmetrically, with both including nitrogen as a component.

## 2.4. Reusability of WO$_3$/MoCl$_5$

The reusability of WO$_3$/MoCl$_5$ was studied via three reaction trials using the same catalyst batch after simple treatment (Figure 9). Once used, the catalyst was dried in the 90 °C oven and was washed with water several times to desorb some dyes that were adsorbed previously. The results show that

there was almost no difference in the efficiency of the catalyst during three reaction times. Due to its high reusability, the catalyst is anticipated to be used as an industrial catalyst.

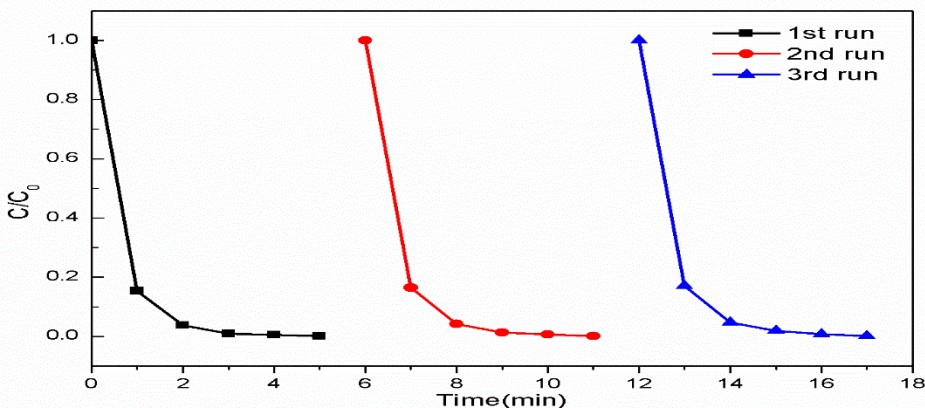

**Figure 9.** Reusability of $WO_3/MoCl_5$ as a catalyst for CV and RB degradation.

## 3. Materials and Methods

### 3.1. Materials

Tungsten(VI) oxide was bought from Sigma Aldrich (Sigma Aldrich Korea, Yongin, Korea) and was dehydrated for 12 h before use. RB and CV were purchased from Sigma Aldrich and were made, respectively, into a 30 ppm and 50 ppm solution with Millipore 18 MΩ pure water. Other chemicals also purchased from Sigma Aldrich, such as toluene, 1,1,3,3-tetramethylguanidine, and hexane were used for precursors or solvents during synthesis and SEM analysis.

### 3.2. Synthesis of the Molybdenum(V) Chloride-Tungsten Oxide Complex

This procedure was followed to synthesize a W-O-Mo complex catalyst. 2 g of $WO_3$ was heated in a Schlenk flask under a vacuum condition for 12 h at 100 °C and was continuously stirred with a magnetic bar throughout the heating. 40 mL of toluene along with 2 mL of 1,1,3,3-tetramethylguanidine was added to the flask and was stirred continuously for 2 h. Toluene (40 mL) was mixed with 2 g of $MoCl_5$ in another Schlenk flask. $N_2$ gas was injected into the flask, and the mixture in the flask was stirred for 4 h. To prevent the Mo from reacting with the atmospheric oxygen, the $MoCl_5$ solution was transferred immediately to the flask containing a $WO_3$ toluene solution. The catalyst solution was stirred for 10 h, vacuum filtered after stirring, and put in the oven at 95 °C. The catalyst was washed using toluene for 10 min, and it was dried with a temperature of 110 °C. Dried heterogeneous $WO_3/MoCl_5$ was inserted into a glass vial and wrapped to prevent contact with the atmospheric oxygen.

### 3.3. Catalyst Characterization

Crystalline structures were collected by X-ray diffraction (XRD, Rigaku model, Miniflex 600 m) with Cu-Kα radiation (λ = 1.5417 Å) at 40 kV, 15 mA (step size 0.02°) in the range of $10° \leq 2\theta \leq 85°$. The morphology of the synthesized material was characterized by transmission electron microscopy (TEM, Hitachi, Tokyo, Japan, H-7600) and scanning electron microscopy (SEM, FEI, Quanta 200) with support of energy dispersive spectra (EDS). For TEM analysis, the catalyst was dispersed in toluene using a sonicator for 3 h and was observed. The catalyst was attached to the carbon tape before observation, and 30.0 kV of electrons were used under an intermediate vacuum state. Concentrations of the dye solutions were determined by UV-Vis absorption spectroscopy (Carry 5000, Varian, Sydney, Australia). To gain the optimum result of UV-Vis spectroscopy, the original dye solution and the dye solution treated with $WO_3/MoCl_5$ were diluted at the same rate until the highest peak was around an absorbance value of 1.0. Fourier-Transform Infrared Spectroscopy (FT-IR, Thermo Fisher Scientific,

Waltham, MA, USA, FT-IR 300) was used to reveal the linkages between the elements of the catalyst. The scan wavelength range was 400–4000 $cm^{-1}$, the step size was 0.5 $cm^{-1}$, and 16 signals were used to clear the noise. Nuclear Magnetic Resonance (NMR, Jeol, Tokyo, Japan, JNM ECP-600 MHz) was used to investigate the difference of the peaks before and after degradation.

### 3.4. Catalytic Tests

The catalytic reaction was conducted in a 15 mL test tube. All dye solutions were made in 1 L volume using purified water with Millipore 18 MΩ. The initial concentrations of CV and RB were both 30 mg × $L^{-1}$ and 50 mg × $L^{-1}$, while the volume was 10 mL for each reaction; 0.01 g of catalysts were employed for each reaction, which is comparably smaller than the quantity of catalysts used in other studies.

The normal light condition was a spontaneous condition, in which there were no highly concentrated light source, such as a UV lamp, directly irradiating the reactor. The only light source that could affect the reaction was a fluorescent light located on the ceiling of the laboratory to light up the whole room. Albeit dim, a bit of light might have influenced the reaction. To exclude at least a portion of the effect of the fluorescent light, the total dark condition was made to verify if the reaction could process without any light. Complete dark was produced by turning off all lights of the laboratory, including lights for the hallway, by conducting the experiment after midnight, and by wrapping the reactor with aluminum foil.

## 4. Conclusions

This research focused on the study of a novel catalyst $WO_3/MoCl_5$ synthesized by thermal synthesis. The degradation reaction of crystal violet dye (CV) and rhodamine B dye (RB) using the synthesized catalyst was studied, including observing the catalyst's reusability. The catalyst reaction was first conducted in a natural environment (normal light condition), without adding a concentrated energy source or using natural light sources, such as sunlight and fluorescent lights that light up the laboratory. Next, the reaction was conducted without any energy source, such as light (dark condition). Both cases gave an identical result for the final percentage of degradation (100%), but the latter case needed more time than the former case. The normal light conditions gave a reaction rate constant of 1.74 $s^{-1}$ on 30 ppm CV, which is superior to any other results from research conducted on RB and CV degradation. Moreover, the time required for 80% degradation for both cases of 50 mg × $L^{-1}$ dye solution was about 1 min, which is extremely rapid. This result was verified for three trials, showing that the catalysts were reusable by simply washing them after the reaction. Moreover, spectroscopic studies revealed that the synthesis progress using the basicity of TMG to deprotonate the hydroxyl hydrogens and adding $MoCl_5$ successfully induced the bondage of W-O-Mo. High-quality $^1$H-NMR spectroscopy concluded that the degradation product of RB treated by pure $MoCl_5$ and $WO_3/MoCl_5$ was different, and careful observation of specific regions offered blueprints for the degradation product.

**Author Contributions:** D.K. designed and supervised the majority of experiments, wrote the manuscript and analyzed the data. H.B. and G.K. designed several experiments and did the experiments with D.K., E.K., D.C., and B.M., N.H.T. provided the concepts and reviewed the paper prior to submission. All authors contributed on paper revision.

**Funding:** This work was supported by the Research & Education Program at the Korea Science Academy of KAIST with funds from the Ministry of Science and ICT.

**Acknowledgments:** All the authors are thankful to Man-Seog Chun, Eun-Young Choi, and Jin Ho Oh for their constant support throughout this research activity.

**Conflicts of Interest:** The authors declare no conflict of interest.

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
