# Peer review of "An External Energy Independent WO3/MoCl5 Nano-Sized Catalyst for the Superior Degradation of Crystal Violet and Rhodamine B Dye"

_catalysts, doi:10.3390/catal9080642_

Round 1

Author Response

Thank you so much for all your review and help.

<Introduction>

1. In the first paragraph, it would be interesting to add information about the human activities that generate these pollutants (crystal violet and rhodamine B), the type of effluents concerned by this kind of pollution as well as typical concentrations of these pollutants in aqueous environment.

è We added additional information about the usages of the pollutant and the more specific details about the effects. We added it as reference of RB as [5] and CV as [10].

2. Lines 44 and 59, the authors refer to "aqueous condition", would it be more appropriate to use the word "phase" instead of "condition" ? .

è We changed the phrase ‘condition’ to ‘phase’.

3. The end of the introduction (from lines 62 to 68) is inappropriate as the authors already give some results and conclusion. To my point of view, the end of the introduction should only present what has been done and what is new compared to the literature already published.

è We removed the repeated part-some results and conclusion and modified the passage to focus on what has been done and what is new compared to the precedented researches.

4. The authors should explain why they choose to mix WO3 and MoCl5 and not other species.

+ They should also refer to the abundant literature on these two catalysts and give the main results concerning them as well as their weaknesses in order to justify their choice and the interest of their study.

 -> We added each explanation that reviewer required and added reference as [25,26,27,28,29,30]

<Results>

1. It sounds strange to begin a paragraph with pictures. I suggest that the authors start the paragraph with the sentence "The morphology of the......in Fig. 1" (line 74) and place Figure 1 below it. Idem for Figures 2 and 3.

-> We have changed the sequence as the reviewer required.

2. Lines 109, 122, 127, "-1" must be put as an exponent after "cm".

-> We have put “-1” as an exponent after cm as the reviewer said.

3. For what is stated Line 109, it could be good to add a reference.

-> We had added the reference as [31], following the reviewer’s request.

4. It is not evident for me looking at Figure 3 about the peak at 975 cm-1 and the weak signals at 565 cm-1 and 450 cm-1.

-> The figure was not big enough and the peaks have not been clear because of it. We enlarged the figure and checked again the parts that reviewer questioned about, and reconfirmed that there is a peak. However, we checked that peak at 24375px-1 is not so big enough to call as prominent peak. Therefore, we eliminated the word prominent. The peak at 11250px-1 might have not been so visible since the IR scan range is to 10000px-1, and it is almost at the end. However, careful observation shows that there is a peak once going down and going up again. It is more easy to observe the peaks if compare the original metal precursor peak and the synthesized catalyst peak.

5. In the catalytic study, the authors should explain how they calculate the reaction rate constants. I understand that the kinetic curves were modeled by a first order kinetic. What are the determination coefficient R² for the fitting between the experimental points and the model? What are the uncertainty on the given values? Were the experiments repeatable?

-> We misused the word determination coefficient R2 as the coefficient of regression. The R2 value at Table 2. (Former Figure 6.) is actually a table including determination coefficient. We added how we calculated the reaction rate constants. The experiments were repeatable. We averaged 3 trials of experiment, and each trials showed almost the similar results. Based on the difference of each experiment, we calculated the uncertainty values of K (rate constant), it showed 0.0007 in maximum, so we used 3 digits under period.

6. In Figure 6, that is not a figure but a table, the unit of the first order kinetic constants must be given in the caption and precision with 4 digits after the comma is not necessary, especially since the uncertainty on the values has not been calculated. The presentation of the table should be changed for the reader to better understand what is indicated in it. As far as I know, R² is not a coefficient of regression but a determination coefficient.

-> We changed figure 6 into table 2. The unit of the first order kinetic constants are now in the caption, as [sec-1]. We reduced the digits into 3 as the reviewer commented that 4 digits are unnecessary. We checked that the calculated values R2 at our figure was determination coefficient as the reviewer said. Thanks for the correction. We added brief explanation about the table, what does each value represent and how they are calculated.

7. Line 145, the authors should give some references from the literature to argue what they stated (reaction rate constants incomparably highest in the range of author's knowledge). It is necessary to give details on the studies that are used for comparison (which concentrations, which catalysts, shape of the reactor, etc.). In paragraphs 2.1 and 2.2, the authors compare WO3 and WO3/ MoCl5 catalysts. Why don't they also compare the efficiency of these two catalysts in the section dealing with the catalytic study?

-> We added details for references as the reviewer requested. We also found researches that used WO3 as a photocatalyst and compared the rate constants. Even if the pure WO3 is used, the catalytic efficiency of WO3 changes by its morphology and particle size. We found 3 references all with different shapes and particle size.

8. Finally, the results dealing with the characterization of the new synthesized catalysts are not used to explain why they permit a quite better efficiency than other catalysts and why they act without needing an external energy source. So that the beneficial impact of the article remains quite low.

-> All of the papers that came out for years were about photocatalysts and the mechanism is well known and easy to investigate. However, our study found a catalyst, which works without any light and there is no background references that we could get help. Also, because this is an irregular phenomenon, there is need to communicate about this catalyst and tell it to the public asap, that many others can start study why it works. Finding the mechanism should be based on more advanced studies and spectroscopy analysis, and that is too much for one paper. We are working on many other spectroscopies and computational estimations, and we will publish the data as a different paper. All of the spectroscopies done in this paper is fully analyzed and all the details that could be inferred are already stated. It is impossible to infer the mechanism or explain why this catalyst has high efficiency by using current spectroscopy that we have done. Also, since this catalyst has at least 50 times higher speed and efficiency than any other catalysts, there is no worry on impact of the article. The phenomenon itself has power of impact and we will subsequently study about the mechanism about the phenomenon. This paper already deserves attention by its energy-free and extremely fast reaction rate.

<Materials and Methods>

1. To my point of view, some information are missing especially in terms of irradiance received by the catalysts when experiment are carried out under normal light. What is the wavelength range? What is the level of irradiation emitted by the lamp and received by the catalyst?

-> Normal light condition in this script is not same as other papers. This is not a photocatalyst. Other papers usually use concentrated light like lamps and irradiate it to the reactor. However, the light that we are talking about is the light that is at the ceiling to light up the room. We need sight to see our reactor and add catalysts to the reactor. The light is a fluorescent light that is on the ceiling. We already wrote down about it in the script. It is impossible to calculate how much lamp has emitted and how much catalyst had received. Our main concept of experiment was that we just did the experiment on ordinary light condition and next did the total dark condition by wrapping the reactor, doing the experiment after 12 at night, etc. How much light the ceiling emits and how much the reactor receives is both not important and impossible to calculate. Please read the experimental part again carefully about the condition.

2. The BET surface area of the catalyst should also have been determined as the synthesized catalysts appear to be quite efficient in terms of adsorption capacity.

-> As we stated our plan, we are working on other spectroscopy to find out the mechanism. Since lines of research is still going on and BET is one of the important part of the paper, we could not reveal it in this paper because same data could not published twice. However, for information, actually the synthesized catalyst has smaller BET surface area than the pristine metal precursor which means that itself is not good adsorbent than the metal precursor. This is one of our main ideas in the next paper, we could not reveal anymore. I hope the reviewer would please understand.

Reviewer 2 Report

The authors describe the use of a WO3/MoCl5 Nano-size catalyst, which degrades crystal violet and Rhodamine B dyes in normal light conditions and without any energy source, with a complete percentage of degradation in very short time.

This work furnishes an interesting contribution, however the manuscript is suitable for publication after minor revisions specified below:

 1.      Pg 3 line 92: the role of TMG is not well specified and should be clarified.

 2.      Pg 3 Figure 2: the figure (a) is unclear and the corresponding table is not formatted appropriately.

 3.      Pg 6 line 140: a reference concerning the RB degradation should be added:

 Science of Advanced Materials, 2014, 6, 1668-1675.

 4.      Pg 8 lines 184-187: the sentence is redundant and could be eliminated from the text.

 5.      Pg 8 Figure 6: it is not appropriately formatted and it should be indicated as Table 1.

 6.      Pg 8 Figure 7: the figure (a) is unclear and should be magnified.

 7.      Pg 9 and 10 Figure 8: the horizontal scale of the NMR spectra should be magnified.

Author Response

Thank you so much for all of your time, effort to give us help and review.

1.      Pg 3 line 92: the role of TMG is not well specified and should be clarified.

è Actually the role of TMG is specified in the experimental session where it explains the synthesis of WO3/MoCl5. However, it would be convenient for the readers if we repeat it again in the session, where the reviewer asked us to do. So we addressed the role of TMG again and related it with the results of EDS analysis.

 2.      Pg 3 Figure 2: the figure (a) is unclear and the corresponding table is not formatted appropriately.

è We magnified figure (a) to make it clear and we renamed it as Figure 2. We changed Figure 2(b) into a chart to an appropriate format and named it as Table 1, since it is actually a table.

 3.      Pg 6 line 140: a reference concerning the RB degradation should be added:

 Science of Advanced Materials, 2014, 6, 1668-1675.

è We had fully understood the reference, and added it as one of the reference as [41]

 4.      Pg 8 lines 184-187: the sentence is redundant and could be eliminated from the text.

è We had eliminated the sentence and modified the sentence no. 188 a bit, only in structural manner to make the manuscript fluent.

 5.      Pg 8 Figure 6: it is not appropriately formatted and it should be indicated as Table 1.

è We had changed the format to an appropriate one and the table is indicated as Table 2., since figure 2(b) has named as Table 1.

 6.      Pg 8 Figure 7: the figure (a) is unclear and should be magnified.

è We had magnified the figure into appropriate size. The figure is now clear.

 7.      Pg 9 and 10 Figure 8: the horizontal scale of the NMR spectra should be magnified.

è We had magnified the figure into appropriate size.
